# The Role of miRNAs in the Development, Proliferation, and Progression of Endometrial Cancer

**DOI:** 10.3390/ijms241411489

**Published:** 2023-07-15

**Authors:** Anna Bogaczyk, Izabela Zawlik, Tomasz Zuzak, Marta Kluz, Natalia Potocka, Tomasz Kluz

**Affiliations:** 1Department of Gynecology, Gynecology Oncology and Obstetrics, Fryderyk Chopin University Hospital, F.Szopena 2, 35-055 Rzeszow, Poland; annabogaczyk@interia.pl (A.B.); tomasz.zuzak@gmail.com (T.Z.); jtkluz@interia.pl (T.K.); 2Laboratory of Molecular Biology, Centre for Innovative Research in Medical and Natural Sciences, Medical College of Rzeszow University, Warzywna 1a, 35-959 Rzeszow, Poland; izazawlik@gmail.com; 3Institute of Medical Sciences, Medical College of Rzeszow University, Kopisto 2a, 35-959 Rzeszow, Poland; 4Department of Pathology, Fryderyk Chopin University Hospital, F.Szopena 2, 35-055 Rzeszow, Poland; marta.kluz@interia.pl

**Keywords:** miRNA, endometrial cancer, epigenetics, carcinogenesis

## Abstract

Endometrial cancer is one of the most common cancers in developing and developed countries. Although the detection of this cancer is high at the early stages, there is still a lack of markers to monitor the disease, its recurrence, and metastasis. MiRNAs are in charge of the post-transcriptional regulation of genes responsible for the most important biological processes, which is why they are increasingly used as biomarkers in many types of cancer. Many studies have demonstrated the influence of miRNAs on the processes related to carcinogenesis. The characteristics of miRNA expression profiles in endometrial cancer will allow their use as diagnostic and prognostic biomarkers. This paper focuses on the discussion of selected miRNAs based on the literature and their role in the development of endometrial cancer.

## 1. Introduction

### 1.1. Endometrial Cancer

Endometrial cancer (EC) was diagnosed in 417,367 women worldwide in 2020, with the highest burden of the disease recorded in North America and Western Europe. The incidence of EC is rapidly increasing. As of 2020, uterine cancer is the fourth most common female cancer in Europe, with an incidence of 12.9–20.2 per 100,000 women and a mortality rate of 2.0–3.7 per 100,000 women [1]. The high incidence rate in North America and Western Europe can be attributed to the high prevalence of lifestyle risk factors for EC, such as high standard of living, aging population, and obesity, which are associated with approximately 50% of EC cases [2].

In the historical morphological division (according to Bokhman’s dualistic theory), EC was classified as type I, the so-called endometroid, which is associated with excessive estrogen stimulation, develops on the basis of endometrial hyperplasia, is more common, and has a favorable prognosis. Type II (non-endometrioid) unrelated to estrogen stimulation has a poor prognosis. Type I includes stage I or II endometrioid adenocarcinoma, while type II EC includes stage III endometrioid adenocarcinoma, serous, clear cell, undifferentiated, and carcinoma [3]. The Cancer Genome Atlas (TCGA), introduced molecular profiling in 2013, which indicates a paradigm shift from morphological to molecular classification [4]. The TCGA studies identified four molecular subgroups characterized by the *POLE* mutation (*POLEmut* group), microsatellite instability (MSI group), which arises from MMRD, high somatic copy number changes (driven by the *TP53* mutation, also called p53abn group), and a low number of copies without a specific molecular profile (NSMP group), each with a separate prognosis [4]. *POLEmut* tumors, despite their aggressive appearance, have an extremely favorable prognosis, while the group with a high copy number driven by the *TP53* mutation has an unfavorable prognosis. The prognosis of tumors with a mismatch repair deficiency (MMRd) and those without a specific molecular profile (NSMP) is relatively favorable [5,6].

The basic treatment of endometrial cancer is surgery and, possibly, subsequent chemotherapy, radiotherapy, and chemoradiotherapy [5,7,8]. The risk of endometrial cancer recurrence is also present in very low-risk cases and is 2.9% within the first 3 years after the end of treatment [9]. Therefore, it is necessary to study the molecular mechanisms in the pathogenesis of endometrial cancer in order to discover new therapeutic methods.

### 1.2. MicroRNAs

The non-coding molecules play a particular role in the regulation of gene expression. The group of regulatory non-coding RNAs includes transport RNA (tRNA), ribosomal RNA (rRNA), antisense RNA (asRNA), microRNA (miRNA), small nuclear RNA (snRNA), small nucleolar RNA (snoRNA), competing endogenous RNA (ceRNA), and piwi-interactive RNA (piRNA) [10].

MiRNAs are a particularly interesting group in terms of regulation of gene expression. They were discovered in 1993 and are non-coding, single-stranded, small RNA molecules about 19–25 nucleotides long. The first ones to be described were small RNA molecules encoded by the lin-4 gene, which regulates the expression of the lin-14 protein in Caenorhabditis elegans by Lee et al. [11].

MiRNA formation begins in the cell nucleus where polymerase II (Pol II) transcribes the pri-miRNA. Pri-miRNA is trimmed by the DROSHA complex and DGCR8 proteins to pre-miRNA. Then, the pre-miRNA is exported by Exportin 5 to the cytoplasm [12]. In this transport, Exportin 5 interacts with the Ran protein. In the cytoplasm, a miRNA duplex is formed from the pre-miRNA, which then separates into two mature single-stranded miRNAs. This process takes place with the participation of DICER and Argonaute 2 (AGO2) [13,14].

MiRNAs function as components of a ribonucleoprotein complex called miRISC (microRNA-induced silencing complex) [15]. Mature miRNA molecules, embedded in miRISC complexes, have the ability to bind to the 3′ untranslated regions (3′UTR) of the mRNA of the target gene. As a result of full nucleotide complementarity, they can lead to transcript degradation. In most cases, miRNAs are usually imperfectly complementary to their target gene and modulate the effect on gene expression via translational repression [16]. The mechanism of action of miRNAs involves binding to a sequence within the RNA-induced silencing complex (RISC), and then gene regulation through translational repression, mRNA degradation, poly(A) tail shortening, and removal of the 5′7-methylguanyl cap [17].

MiRNAs are involved in various cellular functions including proliferation, migration, invasion, and the epithelial–mesenchymal transition (EMT) process. EMT is an important process where epithelial cells lose cell–cell contact and undergo a gradual transformation from an epithelial to a mesenchymal phenotype, which includes, i.e., cytoskeletal remodeling and migratory activity [18]. MiRNAs affect genome instability, regulate metabolism, and influence the apoptosis process of tumor cells; in addition, they also play a role in angiogenesis and immune escape of cancer [19,20,21,22,23]. They may also regulate gene expression within the cell or may be released outside the cell. This leads to the regulation of gene expression in neighboring cells. Therefore, they are regulators of a complex network of processes occurring in the tumor microenvironment [24]. For instance, the let-7 family acts as a regulator of normal cell differentiation and proliferation and inhibits the growth of cancer cells. Let-7 levels are crucial for development cells and act directly on *RAS* genes via LIN28 [25]. Masood N. et al. have reported mutual inhibition of let-7 and LIN28, but let-7 also inhibits IL-6 in embryonic cells, resulting in high levels of NFKB. NFKB together with c-Myc has a stimulating effect on the formation of higher levels of LIN28 in cells. This increase in LIN28 then leads to a marked decrease in let-7 [26].

During carcinogenesis, the miRNA expression profile is significantly dysregulated. This is the result of many changes, including amplification and deletion of genes or epigenetic abnormalities. Moreover, miRNA expression is deregulated in cancer as a result of defects in their biogenesis machinery, including DICER and DROSHA [17,27]. Overexpressed miRNAs in cancers can function as oncogenes and promote cancer development through downregulated tumor suppressor genes or genes that control cell differentiation or apoptosis. Underexpressed miRNAs can function as cancer suppressor genes and can inhibit cancers by regulating oncogenes or genes that control cell differentiation or apoptosis [28]. Such examples are miR-181a, miR-181b, and miR-181c, which are downregulated in glioma [29], while miR-181a and miR-181b are overexpressed in patients with acute lymphoblastic leukemia (ALL) [30].

In addition, they are involved in the regulation of cancer-related signaling pathways, including the JAK/STAT3 transcription pathway [31], the NF-KB pathway [32], and the MAPK/ERK pathway [33]. They may also affect other miRNAs and may be subject to mutual regulation of miRNAs: miRNAs [34].

MiRNAs can be regulators of the above processes, but they can also be regulated by such molecules as circular RNAs, long ncRNAs, or pseudogenes. CircRNA molecules act as ”sponges” for miRNA and thus regulate the amount of free miRNA. They are post-transcriptional regulators of gene expression regulation. A single circRNA molecule can bind to several miRNAs [35,36].

Currently, miRNAs are attractive candidates for therapeutic targets in the treatment of malignancies. Therefore, identifying their targets is essential for cancer research. They are used to assess response to treatment. MiRNAs have also been found to induce chemoresistance in various cancers [37]. A relationship has also been found between miRNA expression and response to treatment, for example, in breast cancer, miRNA-205 was upregulated in tamoxifen resistance cells MCF-7/TAMR-1 (M/T) and M/T cell-derived exosomes (M/ T-Exo) [38]. In lung cancer, the relationship between miRNA levels and cisplatin resistance has also been demonstrated, and miR33b-3p, miR-425-3p, miR-124, miR-295-5p are overexpressed, while miR-98, miR-26a, miR-107 or miR-17 [39]. In advanced colorectal cancer, resistance to FOLFOX (5-fluorouracil, leucovorin, and oxaliplatin) has been shown to correlate with miR-19a overexpression [40]. A similar situation occurs when treating patients with advanced CRC with anti-VEGF or anti-EGFR inhibitors, e.g., overexpression of miR-126 has been correlated with resistance to bevacizumab [41].

The phenomenon of chemoresistance also occurs in endometrial cancer. MiR-222-3p has been shown to increase raloxifene resistance by suppressing Erα expression in cancer cells. MiR-222-3p may be a potential target for restoring ERα expression and response to antiestrogen therapy in the EC. With the upregulation of miR-222-3p, RL95-2 cells were less sensitive to raloxifene. In contrast, AN3CA cells were more sensitive after miR-222-3p inhibition [42].

An interesting direction of research is resistance to cisplatin. Cisplatin has been used in the treatment of various cancers as an effective chemotherapeutic agent for several decades. Wang et al. showed that overexpression of miR-135a increased the survival of endometrial cancer cells after cisplatin treatment. And the decrease in miR-135a expression reduced the survival of endometrial cancer cells after cisplatin treatment. Researchers indicated that miR-135a regulated cisplatin resistance in EC cells. The expression level of miR-135a was associated with cisplatin-induced apoptosis in EC cells. These findings suggest that miR-135a may affect the chemosensitivity of endometrial cancer cells to cisplatin treatment [43].

The most commonly used material for miRNA detection is tissue obtained during surgery. They can also be detected in blood, serum, urine, and other body fluids [44,45]. The method of using blood collection instead of abrasion of the uterus is much easier and carries a lower risk of complications, such as uterine infection. In the future, miRNA profile analysis may be included in routine blood tests for endometrial cancer screening in the general population. This is of great importance, especially for patients living in places with difficult access to health care.

## 2. MicroRNAs in Endometrial Cancer Patients

### 2.1. The Process of Carcinogenesis

The development of endometrial cancer is a complex process involving multiple oncogenes and tumor suppressor genes, although the molecular mechanisms are not clear. In recent years, many studies have been conducted on the expression and function of miRNAs in endometrial cancer [46,47,48,49]. Cancer progression involves several key steps (Figure 1), including primary tumor growth, migration, and local invasion, transendothelial migration of cancer cells into vessels known as intravasation, survival in the circulatory system, extravasation, and niche formation (pre-metastatic niche). This is followed by the recruitment of tumor-promoting immune cells and metastasis. Each stage of carcinogenesis is regulated by many miRs (Table 1).

### 2.2. Risk Factors and Prognostic Factors

Risk factors for EC include genetic predisposition to Lynch syndrome and Cowden syndrome, polycystic ovary syndrome (PCOS), use of tamoxifen, infertility, diabetes, and obesity [99,100,101]. On the other hand, prognostic factors for EC include the patient’s age, stage of endometrial cancer, involvement of lymph nodes, and lymphatic space (LVSI) [102].

#### 2.2.1. Polycystic Ovary Syndrome (PCOS)

PCOS is the most common endocrine disorder among young women of reproductive age. It is characterized by rare or absent ovulation and hyperandrogenism. In patients with PCOS, other factors of endometrial cancer are often diagnosed, such as diabetes, obesity, and nulliparous status [103]. MiRNAs have been studied in patients with polycystic ovary syndrome (PCOS) [104]. This is a large group of mostly young women. The worldwide prevalence of PCOS ranges from 4–21% [105]. Obesity with or without concomitant diabetes often coexists in these women [106]. Many abnormalities and overexpression of many miRNAs have been found in them. It has been discovered that changes in their expression occurring in PCOS are also often associated with metabolic syndrome, which includes hypertension, dyslipidemia, central obesity, and impaired glucose tolerance [107].

PCOS increases the risk of developing endometrial cancer 2.7 times. It has been shown that women with PCOS also change the level of miRNAs, e.g., miR-27a-5p, the level of which is increased in serum-derived exosomes. MiR-27a-5p plays a role in EC cell migration and invasion by regulating SMAD4 [108].

#### 2.2.2. Obesity and Diabetes

Altered expression patterns of miRNAs are not only associated with cancer development but also with comorbidities that are common in patients with EC. These diseases include obesity, type 2 diabetes, and cardiovascular diseases [109]. They are risk factors for many cancers, including endometrial cancer. Another important risk factor in EC carcinogenesis is excessive estrogenic stimulation of the endometrium with a simultaneous lack of progesterone effect. This is the case with polycyclic ovary syndrome (PCOS), obesity, functional tumors, and iatrogenic use of estrogens [103,110,111]. Serum miRNA levels were abnormal in obese women or women with type 2 diabetes, data are summarized in Table 2.

#### 2.2.3. Aging of the Body

Aging is a natural and multifactorial phenomenon characterized by the accumulation of degenerative processes, which in turn are underpinned by multiple changes and damages in molecular pathways [119,120]. Despite many theories that have been proposed to explain the phenomenon of aging, none has been able to fully explain the mechanisms that drive the underlying process so far [120]. MiRNA expression also changes with the age of patients [121]. Some of these were downregulated in long-lived individuals, such as let-7, miR-17, and miR-34 (known as longevity miRNAs). They are conserved in humans and probably promote life extension. Conversely, they are upregulated in age-related diseases such as cancer [122]. MiR-151a-3p, miR-181a-5p, and miR-1248 are downregulated with age [120,123]. In contrast, miR-21 and miR-23a expression increases in middle-aged humans and decreases in advanced age [124].

#### 2.2.4. Involvement of the Lymph Nodes Metastasis

One of the most important prognostic factors used to determine the stage of EC and possible adjuvant treatment is the presence of neoplastic cells in the lymph nodes. Lymphadenectomy is associated with significant surgical and postoperative risks. The use of sentinel lymph node mapping (SLNM) has emerged as an alternative method for total lymphadenectomy in the EC [125]. However, controversy remains over the use of SLNM in high-risk diseases and its false-negative rate (3%) [126]. Reliable SLNM mapping requires surgeons and institutions to be equipped with appropriate knowledge and skills. In addition, SLNM mapping is performed during the operation. It is also worth remembering that the involvement of lymph nodes may also engage paraaortic nodes. Isolated involvement of the paraaortic nodes in patients without pelvic nodal metastases was only 1% [127]. Therefore, finding pre-operative methods that can accurately identify LNM (lymph node metastasis) would be of great clinical value. MiRNA mapping may prove to be such a way. A correlation of miRNAs depending on the presence of relapses in lymph nodes has been demonstrated. Table 3 summarizes the above data.

#### 2.2.5. The Impact of miRNA Changes on Survival and Recurrence in Patients with Endometrial Cancer

Mortality related to endometrial cancer continues to increase [136]. Although most patients with endometrial cancer have a tumor confined to the uterus that is treated by hysterectomy with or without adjuvant therapy, the advanced disease has a poor prognosis [137]. Although early-stage endometrial cancer is associated with a favorable 5-year relative survival rate (96%), the rate is only 18% in patients with distant metastases [138]. Patients with an increase in the recurrence-free period were examined and changes in microRNA levels were also found here (Table 4).

On the other hand, the factors associated with shortening the recurrence-free period are a high expression of miR-21 [128] or miR-205 [142].

Changes in miRNA expression levels are clearly visible in tumor tissue but can also be seen in plasma/serum. In patients with endometrial cancer, two groups—increased and decreased expression—were distinguished. Disorders of miRNA expression in plasma/serum are summarized in Table 5. Such studies are particularly important due to the ease of obtaining material for research.

## 3. Overview of Selected microRNAs

### 3.1. MiR-205

Multiple studies have shown that miR-205 is overexpressed in the EC compared to normal endometrial tissues. It was previously reported that miR-205 upregulation was significantly correlated with advanced disease stage, relapse incidence, and poor EC survival rates [142,149]. MiR-205 is involved in regulating the expression of PTEN, which is the most common mutated tumor suppressor gene [150,151]. This mutation is also found in endometrial cancer and accounts for 25–83% of cases [152]. PTEN performs an important inhibitory function by promoting apoptosis and proliferation. Its deletion or mutation leads to carcinogenesis. Zhang et al. observed that miR-205 was significantly upregulated in the Ishikawa cell line compared to normal endometrium [153]. MiR-205 interacted directly with the 3′-UTR region of the PTEN gene. Overexpression of miR-205 decreased PTEN mRNA and protein levels in Ishikawa cells. Zhang et al. further reported that miR-205 blocked PTEN translation and activated the AKT pathway. Constitutive activation of AKT contributes to tumor progression and regulates several downstream targets (e.g., TP53 and BCL-2). Downregulation of miR-205 expression is followed by decreased levels of p53 protein and increased levels of BCL-2 protein. Since the TP53 and BCL-2 genes are involved in cell growth, apoptosis, and proliferation, these results provide the basis for further research into the role of miR-205 in EC cells. Also, the rate of cell apoptosis can be inhibited by miR-205. MiR-205 acts as an oncogene and inhibits cellular apoptosis in the EC by targeting the PTEN/AKT pathway [153].

It should also be noted that miR-205 plays an important role in the migration and invasion of endometrial cancer. This mechanism is based on the targeting of miR-205 to the AKT pathway.

Inhibition of E-cadherin expression and promotion of Snail expression by activating AKT and downregulation of glycogen synthesizing kinase 3β were associated with overexpression of miR-205. The molecular mechanism of action of miR-205 regulating the epithelial–mesenchymal transition (EMT) by activating AKT signaling in endometrial cancer cells in the HEC-50B and HEC-1-A cell lines was described by Jin C. et al. [154].

In addition, several studies have reported that miR-205 inhibits the tumor suppressor gene JPH4, promoting tumorigenesis and progression [46].

### 3.2. MiR-34

The miR-34 family has three members, i.e., miR-34a, miR-34b, and miR-34c. MiR-34a, b, and c are encoded by two different transcription units. MiR-34a is located on chromosome 1p36.22 and has a unique transcript, while miR-34b and miR-34c share one transcript that is located on chromosome 11q23.1 [155].

MiR-34 is a direct target of the tumor protein p53 (TP53), a tumor suppressor gene that causes cell cycle arrest and apoptosis when activated under cellular stress. Inactivation of p53 can result in a cellular environment that contributes to oncogenesis [52].

It was shown that miR-34c acted as a tumor suppressor in human endometrial carcinoma 1b (HEC-1b) with the E2F3 transcription factor being one of its targets [156].

Reduced miR-34 expression is a negative prognostic factor for serous endometrial cancer and is strongly associated with LVSI. These data reinforce knowledge about the miR-34 family (miR-34a, b, and c), which appears to act as a tumor suppressor [157].

The miR-34 family acts as a negative regulator of cancer-related EMT and plays a large role in suppressing carcinogenesis and delaying tumor progression. As an excellent tumor suppressor, miR-34a is a cancer therapy agent. Many studies have been conducted on miR-34a and verified its suppressive role in cancer. However, some challenges have arisen with the use of miR-34a therapy. One of them is the aforementioned miRNA degradation. RNase is rich in serum and easily denatures miR-34a, as a result of which miR-34 cannot penetrate the capillary endothelium and does not reach its target cells [158].

### 3.3. MiR-21

MiR-21 is overexpressed in almost all human cancers. It acts as an oncogene and may be a useful clinical biomarker and therapeutic target. Its level also increases in endometrial cancer [128].

Researchers have demonstrated different mechanisms of action of miR-21. Yan et al. studied its oncogenic role by inhibiting the tumor suppressor gene FBXO11 (a member of the F-box subfamily lacking a distinct unifying domain), subsequently inhibiting apoptosis and stopping protein degradation [78]. In another study (Tu et al.), the opposite effect of miR-21 and GAS5 (growth arrest-specific transcript 5) was observed. Reduced expression of GAS5 in tumor-associated macrophages (TAMs) in endometrial cancer has been observed. Its anticancer role consists in promoting phagocytosis, presenting antigens, and activating cytotoxic T lymphocytes. MiR-21 as an oncogene inhibits the suppressive effect of GAS5 in endometrial cancer cells [159].

As mentioned earlier, miRNAs bind to target mRNAs through sequence complementarity and lead to inhibition of translation and mRNA destabilization. It is known that this process can be influenced by lncRNAs, through lncRNA:miRNA interactions. An example of a lncRNA is MEG3, which changes the expression of miR-21 [160,161].

Another mechanism of miR-21 involvement was investigated by Li Xiao et al. and is related to tumor cell hypoxia [162].

Under hypoxic conditions, cancer cells produced significantly more exosomes than cells in normoxic conditions. Hypoxia increased miR-21 expression in exosomes. Monocytes were also transformed into M2-like polarizing macrophages by delivery of exomal miRNA-21, which may be a mechanism for the immune escape of tumor cells. MiR-21 may induce a potential mechanism for creating an immune microenvironment in endometrial cancer progression [162]. Hypoxia is, therefore, an aggressive feature of endometrial cancer and an increase in miR-21 levels. This increase results in the downregulation of PTEN and a strong increase in L1CAM gene expression to promote cancer cell invasion and metastasis [162].

Overexpression of miR-21-5p has also been reported to promote epithelial-to-mesenchymal transition (EMT). In contrast, miR-21-5p silencing reversed EMT in endometrial cancer cell lines. This mechanism is related to the SRY-box 17 (SOX17). Overexpression of miR-21-5p significantly inhibits SOX17 protein expression in endometrial cancer cell lines. SOX17 has a suppressive effect and its overexpression promoted mesenchymal to epithelial transition, while SOX17 silencing induced EMT in endometrial cancer cell lines [62].

MiRNAs also have their regulators. Such a regulator for miR-21 is circRNA, i.e., circFAT1. Wu et al. investigated this relationship and described CircFAT1, which was upregulated in EC and positively correlated with miR-21 in EC tissues. In RL95-2 and HEC-1-A cells, circFAT1 overexpression increased miR-21 expression and decreased miR-21 gene methylation, while miR-21 overexpression did not alter circFAT1 expression. Through stem cell analysis, it was shown that overexpression of circFAT1 and miR-21 had an effect on the number of stem cells that increased. In contrast, miR-21 inhibition resulted in a reduction in the number of stem cells. In addition, the miR-21 inhibitor suppressed the role of circFAT1. In conclusion, circFAT1 is upregulated in the EC and can increase the number of tumor cells by upregulating miR-21 [163].

### 3.4. MiR-182

It promotes cell proliferation by targeting the tumor suppressor gene TCEAL7 (transcription elongation factor 7-like), which interacts with the E-box sequences of the Myc cyclin D1 target gene. By affecting the regulation of Myc activity and the expression of cyclin D1, it causes cell proliferation and malignant transformation. Downregulation of TCEAL7 is associated with larger tumor size, higher tumor stage, and poor prognosis [164].

According to Donkers, miR-182 can potentially be used to distinguish high-grade disease from low-grade disease [46].

Devor et al. studied the association of miR-182 and the altered expression of cullin-5(CUL5), a member of the ubiquitin ligase family cullin-RING E3. They showed that there were two miR-182 binding sites in the 3′-UTR of the CUL5 gene and that miR-182 was overexpressed in two EC model cell lines of Ishikawa H and Hec50co. They showed that CUL5 was the target of miR-182 in EC. Upregulation of miR-182 results in downregulation of CUL5, which promotes EC proliferation [165].

Myatt et al. studied changes in the levels of some miRNAs and the tumor suppressor FOXO1. They showed that FOXO1 was downregulated in endometrial cancer compared to normal endometrium. Whereas, the miRNA including miR-182 was upregulated. The target of miR-182 was probably the 3′-untranslated region of FOXO1 transcripts [166].

### 3.5. MiR-200

It is a whole family consisting of miR-200a, miR-200b, miR-200c, miR-429, and miR-141. It negatively regulates two transcription factors, ZEB1 (Zinc finger E-box-binding homeobox 1) and ZEB2, which are well-known suppressors of E-cadherin transcription [55,167]. E-cadherin is a calcium-dependent transmembrane epithelial adhesive molecule involved in cell cohesion. Reduced expression of E-cadherin has been linked to reduced cell–cell adhesion, metastasis potential, tumor differentiation, and deep myometrial invasion in endometrial and other cancers. In endometrial cancer, loss of E-cadherin is strongly associated with histological subtypes where loss is more prevalent in EEC grade 3 compared to serous carcinoma [46].

By targeting E-cadherin transcriptional repressors ZEB1 and ZEB2, the miR-200 family can regulate the epithelial-to-mesenchymal transition and protect cancer cells from apoptosis [168].

MMP2 is an enzyme that degrades type 4 collagen, the main structural component of basement membranes. This enzyme plays a role in the menstrual breakdown of the endometrium, regulation of vascularity, and tumor metastasis. The natural MMP2 inhibitor is TIMP2 (tissue metalloproteinase 2 inhibitor), it is a metastasis suppressor. MiR-200b suppresses TIMP2 expression and increases the activity of matrix metallopeptidase 2 (MMP2) [168].

MMP2 expression in endometrial cancer correlates with the histological grade of the tumor, its invasion, or metastases. Increased MMP2 expression and low TIMP2 expression are the strongest markers of endometrial malignancies with a high risk of local and distant metastases [168].

Both miR-200a and miR-200b belong to the miR-200 family, but they have different target genes, which is related to the difference in the seed regions [168].

MiR-200b is overexpressed in endometrial adenocarcinomas. It specifically inhibits TIMP2 expression and increases MMP2 activity in HEC-1A cells. Both MiR-200b and TIMP2 and MMP2 probably play an important role in the initiation and further development of endometrial adenocarcinoma [168].

### 3.6. MiR-103

MiR-103 has an oncogenic effect. Overexpression of miR-103 increases EC cell proliferation, while downregulation has the opposite effect. ZO-1 is directly suppressed by miR-103. Silencing ZO-1 significantly promotes EC cell proliferation [67].

A possible therapeutic target for miR-103 is TIMP-3 (tissue inhibitor of metalloproteinase 3 expression). Yu et al. studied the effect of miR-103 on TIMP using a TIMP-3 inhibitor. They showed that miR-103 after transcription downregulated the expression of the tumor suppressor TIMP-3 and stimulated growth and invasion in endometrial cancer cell lines [169].

An important pathway in the pathogenesis of endometrial cancer is the GAS5-miR-103-PTEN pathway. GAS5 is a tumor suppressor gene important in stopping cancer formation. It works by inhibiting the expression of the miR-103 oncogene, which increases the expression of PTEN and promotes the apoptosis of cancer cells. Guo et al. conducted a study in which they studied the GAS5-miR-103-PTEN pathway and found that it may be a new therapeutic target in the treatment of endometrial cancer [170].

### 3.7. MiR-105

The miR-105 family, which consists of three members (miR-105-1, miR-105-2, and miR-767). It is located on the human Xq28 chromosome. MiR-105 may play an oncogenic or suppressor role in various cancers [171]. In endometrial cancer, it serves as a tumor inhibitor. It is weakly expressed in tumor tissues and endometrial cancer cell lines. Further upregulation of miR-105 inhibits the proliferation and metastatic potential of endometrial cancer cells. A potential target of miR-105 is SOX9. It has been validated as an miR-105 target transcript. MiR-105 likely inhibits the epithelial–mesenchymal transition and gastric cancer metastasis by targeting SOX9. SOX9 is well established as a redundant transcription factor regulating many developmental signaling pathways and its aberrant expression has been associated with tumor initiation, proliferation, metastasis, and stem cell maintenance. This is also true for endometrial cancer, as SOX9 has been reported as an independent risk factor for endometrial hyperplasia in the uterine epithelium, which is a precursor to endometrial cancer. In addition, overexpression of SOX9 was found to cause proliferation of endometrial cancer cells. Other molecules can also act through miR-105, such as Circ_0109046, which acts as an oncogene in endometrial cancer. This affects the development of cancer and its metastases [172].

In other cancers, its function in promoting tumor metastasis has been demonstrated by destroying the vascular endothelial barrier. This mechanism is based on the ZO1 protein and has been described in breast cancer [79].

### 3.8. MiR-136

MiR-136 has also been studied in various cancers. MiR-136 has been identified as a tumor suppressor gene in various adenocarcinomas such as breast cancer, colon cancer, and lung cancer [173,174,175].

MiR-136 acts as spongeRNA for circ_PUM1. Circ_PUM1 plays a key role in the development and progression of endometrial cancer, mainly through the uptake of miR-136 via a ”sponge” effect, thereby promoting the expression of the NOTCH3 target gene [176].

Zong et al. revealed that miR-136 was an anti-proliferative and anti-metastatic miR in the EC [176].

MiR-136 acts as a spongeRNA also for another circRNA, i.e., circ_0109046. Shi Y. et al. studied the mechanism of action of circ_0109046 sponged miR-136 to regulate HMGA2 via the ceRNA mechanism [177]. Shi Y et al. showed that miR-136 directly targeted HMGA2. HMGA2 is a common oncogenic factor and is involved in various cellular processes including cell proliferation, apoptosis, and differentiation. Meanwhile, HMGA2 accelerates tumor progression in gynecological cancers, including cervical cancer, breast cancer, and ovarian cancer. Ma et al. suggested that miR-302a-5p/367-3p-mediated HMGA2 promoted EC cell malignancy [178]. In a study by Shi Y. et al., HMGA2 was identified as a target for miR-136. Importantly, HMGA2 augmentation abolished miR-136 mimic inhibition in EC development [177].

Li et al. studied miR-136 levels in EC stem cells. They showed that it was significantly reduced in EC tissues and its expression correlated with different FIGO stages and grades. By means of a survival analysis, it was shown that patients with low miR-136 expression had a worse prognosis. It was also shown that the expression of miR-136 in endometrial cancer stem cells (ECSC) was significantly lower than in non-stem cells [179]. Overexpression of miR-136 can inhibit EC cell proliferation, migration, and invasion. Overexpression of miR-136 may also promote cell apoptosis and G0/G1 cell cycle arrest. They also inhibit the ability of EC cells to form a ball [179].

### 3.9. MiR-155

MiR-155 negatively regulates p38 protein levels by directly binding to the 3′UTR region of the target mRNA. MiR-155 has a negative effect on the functioning of DCs (dendritic cells, antigen-presenting cells), which plays an important role in the activation of anticancer immune responses. It acts by silencing p38, via the p38 MAPK14 pathway [73].

Yamamoto and Imai studied microsatellite instability (MSI), which is a hallmark of Lynch syndrome [180]. This is an important direction of research because the probability of developing EC as the first tumor in this syndrome is 40–60% [181]. They showed that overexpression of miR-155 or miR-21 downregulated MMR gene expression. MSI-induced frameshift mutation gene targets are involved in essential cellular functions including, for example, DNA repair (MSH3 and MSH6), cell signaling (TGFBR2 and ACVR2A), apoptosis (BAX), epigenetic regulation (HDAC2 and ARID1A) and processing miRNAs (TARBP2 and XPO5), and the MSI + CRC subset reportedly shows a mutant phenotype of the miRNA machine [180].

Another study observed reduced levels of IGF1 (insulin-like growth factor 1), MYLK (myosin chain kinase), and overexpression of SOD2 (superoxide dismutase 2) associated with dysregulation of proliferative processes in the EC. MiR-155 was possibly involved in the regulation of MYLK activity in the EC. Its overexpression may promote uncontrolled tumor proliferation and progression [182].

MiR-155 acts on the angiotensin II receptor type 1 (AGTR1) by inhibiting it. Choi et al. studied this compound using anti-miR-155. They found that combination therapy with anti-miR-155 and losartan had a synergistic effect and has an antiproliferative effect [183].

### 3.10. MiR-372

It acts as a tumor suppressor and inhibits the occurrence and development of endometrial cancer. Its expression is much lower in the EC than in healthy endometrial tissues. Overexpression of miR-372 suppressed cell proliferation, migration, and invasion and led to G1 phase arrest. MiR-372 also promotes apoptosis of endometrial cancer cells in vitro. Researchers detected the expression of known miR-372 targets in other malignancies. They showed that cyclin A1 and cyclin-dependent kinase 2 (CDK2) were downregulated by miR-372. It was also shown that transfection of miR-372 reduces the expression of RhoC, matrix metalloproteinase 2 (MMP2), and MMP9, while it increases expression of cleaved polymerase poly (ADP ribose) (PARP) and bcl-2-linked protein X (Bax) [184].

Another tested target for miR-372 was PRMT6 (protein arginine methyltransferases). PRMT6 overexpression promotes EC cell proliferation and migration and is significantly associated with higher tumor histology grades and unfavorable prognosis. PRMT6 induces AKT and mTOR phosphorylation in the EC. MK2206 or rapamycin inhibits the AKT/mTOR pathway via PRMT6. miR-372-3p expression downregulates PRMT6. In clinical trials, PRMT6 expression was associated with low miR-372-3p expression [185].

### 3.11. MiR-93

MiR-93 derives from the paralog (miR-106b-25) of the miR-17-92 [186] cluster. Its high expression is associated with the short survival of patients with endometrial cancer. In biological experiments conducted in vitro, miR-93-5p overexpression has been shown to promote the proliferation and migration of endometrial cancer cells. Therefore, it is very possible that miR-93-5p promotes the development of endometrial cancer [187].

Chen et al. studied miR-93, which was highly expressed in endometrial cancer tissues. Overexpression of miR-93 promoted the migration and invasion of endometrial cancer cells and decreased E-cadherin expression and increased N-cadherin expression without changing RhoC expression in the EC. MiR-93 promotes EMT, migration, and invasion in endometrial cancer cells by regulating FOXA1 [186].

### 3.12. MiR-125

One important family of miRNAs is the miR-125 family, which includes miR-125a, miR-125b1, and 125b2, which produce nearly identical products of different genes. The miR-125b is of particular interest. Human miR-125b is found throughout the human body and is the highest expressed in the brain and ovaries, followed by the thyroid, pituitary, epididymis, spleen, testes, prostate, uterus, placenta, and liver. MiR-125b can be upregulated, e.g., in colorectal cancer or hematopoietic tumors. However, it may be strongly reduced in breast tumors and hepatocellular carcinoma [188]. Shang et al. showed that miR-125b was downregulated by about 30% in endometrial cancer. Downregulation of miR-125b increased cell invasiveness that could be rescued by overexpression of miR-125b. The direct target for miR-125b is ERBB2. ERBB2 has been shown to modulate microRNA activity by binding to microRNA targeting sequences on the 3′UTR of target mRNAs. ERBB2, encoded as a member of the epidermal growth factor (EGF) receptor family of receptor tyrosine kinases, is associated with increased invasion as a proto-oncogene [189].

### 3.13. MiR-222

MiR-222 has been studied as one of the markers to detect endometrial cancer in patients. Montagnana et al. showed higher serum miR-222 levels in EC patients compared to controls [147].

Donkers et al. studied urinary miR levels and showed that miR-222 expression was significantly reduced in older women [45].

Liu and others studied miR-222-3p and showed that it targeted Erα. miR-222-3p expression is negatively correlated with ERα. Overexpression of miR-222-3p in RL95-2 cells promotes cell proliferation, increased invasiveness, and induces a G1 to S phase shift in the cell cycle. In addition, the researchers showed that miR-222-3p expression was significantly lower in ERα-positive than in ERα-negative EC tissue samples and that the miR-222-3p expression level is inversely correlated with Erα expression. The miR-222-3p expression level is lower in lower-grade tumors. In addition, miR-222-3p was positively associated with lymph node metastases [42].

### 3.14. Let-7

The Let-7 miRNA family was first discovered in nematodes and has as many as 13 members that are located on 9 different chromosomes. Their role is to control the time of division, differentiation, and proliferation of stem cells. They play an important role in carcinogenesis. Its reduced expression is found in the bladder, breast, colorectal, cervical, endometrial, head and neck, lung, ovarian, prostate, and kidney cancers. let-7’s role is to regulate cell differentiation and proliferation as well as inhibit cancer cells by directly acting on RAS genes via LIN28. Let-7 and LIN28 both have inhibitory effects on each other. Decreased let-7 levels result in increased RAS levels leading to tumor cell proliferation [26].

Zhang et al. have shown that the activated estrogen receptor can repress BAX expression by regulating a group of microRNAs including but not limited to members of the has-let-7 family. This results in the promotion of an increased BCL2/BAX ratio as well as increased survival and proliferation in affected cells. These ER-regulated has-let-7 microRNAs can be detected in most endometrial hyperplasias and may be potentially useful indicators of estrogen overexposure [58].

Let-7 (lethal-7) has been described in the pathogenesis of ovarian germ cell tumors (GCT). Let-7 is regulated by the RNA-binding protein LIN-28 homolog A (LIN28). GST LIN28-positive tumors have been shown to be downregulated let-7 [190].

### 3.15. MiR-429

MiR-429 belongs to the miR-200 family and its dysregulation is involved in the epithelial–mesenchymal transition (EMT), progression, development, invasion, metastasis, apoptosis, and drug resistance of various cancers.

Its target is *PTEN*, which is a significant tumor suppressor gene in the EC. As reported by Yoneyama et al., overexpression of miR-429 in EC results in downregulation in EWG *PTEN*. The putative recognition site for miR-429 is the 3′ untranslated *PTEN* region (3′-UTR) [129].

## 4. Summary

MiRNAs play an important role in carcinogenesis and have been the subject of many studies in this field. Carcinogenesis is a complex process involving the dysregulation of many genes, and, therefore, studies on the functions and targets of miRNAs will expand our understanding of the molecular mechanisms that control cancer development [168].

The role of miRNA is the subject of research, e.g., in the etiopathogenesis of polycystic ovary syndrome (PCOS). It has been proven in the literature that in patients with metabolic syndrome a change in miRNA expression occurs. The question we are still looking for an answer to is whether these changes are the direct cause of endometrial cancer and if carcinogenesis can be prevented by changes in epigenetics. Another question is how can changes in miRNAs be observed in this group of patients since it is known that their levels can change with the aging of the body.

A large group of PCOS patients uses Metformin [191]. It is a hypoglycaemic drug that, in short, works in three ways: it reduces hepatic glucose production, increases muscle insulin sensitivity, and delays glucose absorption in the intestine [191]. It also affects changes in miRNA expression [192]. MiR-20a-5p expression was shown to be increased in women with PCOS using Metformin [193]. In contrast, the following miRNAs were decreased: miR-122, miR-29a, and miR-223 [194]. Metformin facilitates weight loss, so will it prevent the development of endometrial cancer?

Studies on the effect of exercise and physical activity on weight loss have also been conducted. A decrease in miR-423-5p expression, a drop in whole-body insulin resistance, and an increase in liver insulin sensitivity have been demonstrated [195]. Some physical exercises can alter miRNAs in skeletal muscle, heart muscle, bone, adipose tissue, liver, brain, and body fluids [196]. However, it has not been shown whether they have an effect on reducing the risk of endometrial cancer.

There is a need to develop new biomarkers for the detection of endometrial cancer and for these markers to accurately distinguish between low-grade (stage 1 and grade 2) or high-grade (stage 3) endometrioid cancer lesions. Chen et al. reported unique miRNA signatures for endometrial cancer, based on the following miRs: miR-652, miR-3170, miR-195, miR-34a, and miR-934 [197]. Such signatures were also created for the differentially expressed miRNA (DEmiR) step to help predict high-risk gynecological cancer patients and demonstrate their role in early and late disease.

MiRNAs may be promising biomarkers. They can be detected in solid tissue samples, but also in blood, serum [44], urine [45], and other body fluids [198]. Since they can be detected in urine and are stable there, urine appears to be a promising non-invasive test for the detection of EC [45,199]. To date, it has not been established which type of sample can be used to obtain the most reliable biomarker for the detection of endometrial cancer.

## Figures and Tables

**Figure 1 ijms-24-11489-f001:**
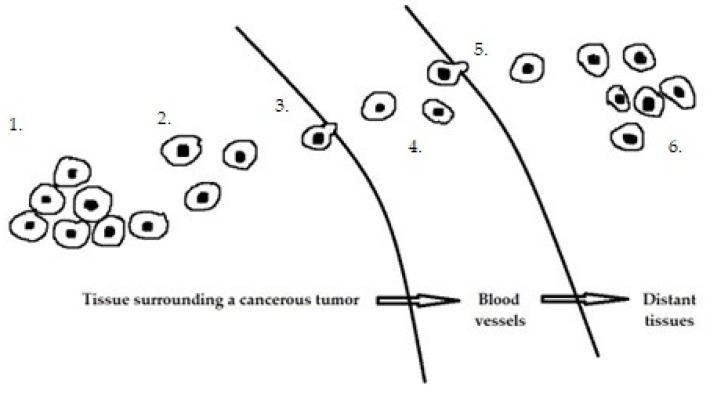
Stages of cancer progression: 1. Primary tumor growth. 2. Migration and local invasion. 3. Transendothelial migration of cancer cells into vessels 4. Survival in the circulatory system. 5. Extravasation. 6. Pre-metastatic niche formation.

**Table 1 ijms-24-11489-t001:** miRNAs of significance for the individual stage of carcinogenesis.

Primary Tumor Growth		
miR-15/16	↓	The miR-15/16 family is a highly expressed tumor suppressor group that targets a large network of genes in T cells to limit their cell cycle, memory formation, and survival. Once activated, miR-15/16 T cells are downregulated, allowing rapid expansion of differentiated effector T cells to mediate a sustained immune response. MiR-15/16 deficiency alters Treg expression of critical functional proteins, including FOXP3, IL2Rα/CD25, CTLA4, PD-1, and IL7Rα/CD127, and results in the accumulation of functionally impaired FOXP3loCD25loCD127hi Tregs [50].
miR-17/91	↑	Involved in immune regulation, three clusters of the miR-17/92 family collectively suppressed IL-12 production in macrophages, and miR-17/92 acts through PTEN to inhibit IL-12 expression by modulating the *PI3K*-Akt-*GSK3* pathway [51].
miR-34	↓	It is involved in the regulation of the cell cycle and apoptosis through p53 signaling [52]. It acts as a tumor suppressor through DNA methylation in both epithelial and hematological malignancies [53].
miR-181a	↑	It can interact with *H3F3B*, *ATM*, *CCDC6*, *TAM15*, *RAS,* and *PLAG1* to promote cell proliferation [54].
miR-200	↓	Targets ZEB1 and blocks the epithelial–mesenchymal transition [55].
miR-211	↑	Targets mRNAs: *POU3F2*, *ZCCHC24*, *PRLR*, *ITPR1*, and *CHRDL1* [56].
miR-222	↑	Targets tumor suppressors *PTEN* and *TIMP3*. Targets *MMP-2* i *MMP-9* [42,57].
Let7	↓	Acting through Lin28, it targets *RAS* genes. Overexpression of let-7 leads to a decrease in RAS production, accelerating the cell cycle, angiogenesis, and cell adhesion. Therefore, under normal conditions, miR let-7 acts as a tumor suppressor gene and inhibits the activation of oncogenes that can lead to the formation of cancer cells [26,58].
**Migration and local invasion**		
miR-9	↑	MiR-9, which is upregulated in breast cancer cells, targets *CDH1*, the mRNA encoding E-cadherin, leading to increased cell motility and invasiveness. The miR-9-mediated downregulation of E-cadherin causes activation of β-catenin signaling, which contributes to the upregulation of growth factor gene expression [59].
miR-10b	↑	It increases invasion, migration, and proliferation and inhibits apoptosis in the EC [60]. It targets *HOXB3* [61].
miR-21	↑	Overexpression of miR-21-5p promoted epithelial to mesenchymal transition. It works through *SOX17* [62].
miR-29c	↓	It affects the expression of HBP1, ITGB1, MCL1, MDM2 and SGK1 [63]. Overexpression of miR-29c reduces COL4A1 production in endometrial cells [64].
miR-34a	↓	Inverse correlation between miR-34a and L1CAM protein expression. A decrease in miR-34a and an increase in L1CAM are associated with poor [65]. MiR-34a is downregulated in endometrial cancer tissues and is negatively correlated with *Notch1* expression [66].
miR-103	↑	Overexpression of miR-103 promotes EC cell proliferation. It works through ZO-1 and triggers its downward adjustments. There is an inverse correlation between ZO-1 and miR-103 [67].
miR-107	↑	MiR-107-5p downregulated Erα mRNA and protein expression [68].
miR-135a	↑	MiR-135a can regulate the epithelial-to-mesenchymal transition (EMT) by altering the expression of E-cadherin and N-cadherin. MiR-135a promotes endometrial cancer cell proliferation by regulating PTEN. Expression levels of PTEN and p-AKT in endometrial cancer cells decreased after miR-135a overexpression [43].
miR-135b	↑	Upregulation of miR-135b significantly reduced FOXO1 protein and mRNA expression, promoting EC proliferation [69].
miR-145	↓	MiR-145 expression is lower in EC tissues than in neighboring tissues. MiR-145 inhibits *SOX11.* MiR-145 targets site 3 (3615) of SOX11 3’UTR to affect *SOX11* expression [70].
miR-148b	↓	Downregulation of miR-148b induced endometrial EMT of the tumor cell as a result of alleviating DNMT1 suppression [71].MiR-148b regulates the expression of endoplasmic reticulum metalloprotease 1 *(ERMP1)* [72].
miR-155	↑	It impairs the functioning of dendritic cells in endometrial cancer which play an important role in the activation of anticancer immune responses. It acts via the p38MAPK14 pathway [73].
miR-214-3p	↓	MiR-214-3p is downregulated and TWIST1 is upregulated in EC tissues and cells. Overexpression of miR-214-3p suppressed migration, invasion, and EMT in EC cells [74].A decrease in miR-214-3p is associated with an increase in NEAT1, HMGA1, and β-catenin [75].
miR-223	↑	MiR-223 modulates the inflammatory response by directly targeting genes mediating signal transduction, including those present in the canonical NF-kB pathway [76].
miR-340	↓	MiR-340-5p is downregulated in the EC compared to adjacent normal tissues. In vitro, miR-340-5p inhibited the migratory capacity of EC cells by downregulating MMP-3 and MMP-9 and prevented TGF-α1-induced EMT by p-eIF4E [77].
**Transendothelial migration of cancer cells into vessels**		
miR-21	↑	It inhibits the suppressive effect of *FBXO11* (a member of the F-box subfamily lacking a clear unifying domain) [78].
miR-105	↑	It targets the ZO-1 protein. In endothelial monolayers, exosome-mediated transfer of tumor-secreted miR-105 effectively disrupts the tight junctions and integrity of these natural barriers to metastasis. Overexpression of miR-105 in non-metastatic cancer cells induces metastasis and vascular permeability in distant organs [79].
miR-126	↓	It is a tumor suppressor and its growth can downregulate VEGF to inhibit EC cell invasion and migration [80].Its decrease correlates with high levels of Lnc-ATB, which induced accelerated tumor growth by regulating the miR-126 PIK3R2 target gene and Sox2-related apoptosis.In the tested RL95 and HEC1A cell lines, the downregulation of Lnc-ATB resulted in the upregulation of miR-126. There was an impairment of cell viability, an increase in caspase-3-related tumor apoptosis, and G1/S arrest [81].
**Survival in the circulatory system**		
miR-26a	↓	Increased peritumoral lymphoid endothelial hyaluronan receptor-1 (LYVE-1) density in LNM patients was negatively associated with the level of miR-26a-5p in primary lesions, indicating that down-expression of miR-26a-5p can induce LNM EC [82].
miR-141	↑	*PPP1R12A* and *PPP1R12B* are targeted and degraded. Both are members of the myosin phosphatase (MYPT) targeting protein family [83].
miR-145-3p	↑	MiR-145 participates in M2 macrophage polarization by targeting IL-16 and upregulating IL-10 [84].
miR-181-a	↑	MiR-181a and miR-181b increased the expression of *PECAM-1* mRNA and protein and VE-cadherin accompanying the differentiation of human embryonic stem cells into vascular endothelial cells [85]. By acting on VE-cadherin, it disrupts the barrier in endothelial cells [86].
miR-424	↓	MiR-424 has a protective role in various types of cancer including endometrial cancer, upregulation of miR-424 inactivated PI3K/AKT signaling mediated by G-1 protein-coupled estrogen receptor (GPER) in endometrial cancer. Moreover, the luciferase report confirmed the targeting reaction between miR-424 and GPER [87].
**Extravasation**		
miR-7	↓	Through the downregulation of the PI3K and MAPK pathways, its dominant role is to inhibit proliferation and survival, stimulate apoptosis, and inhibit migration [88].
miR-21	↑	It inhibits the expression of the *SOX17* protein and promotes epithelial-to-mesenchymal transition (EMT) [62].
miR-31	↓	MiR-31 acts as an oncogene in endometrial cancer by suppressing the hippopotamus pathway. MiR-31 significantly suppressed mRNA luciferase activity in conjunction with the LATS2 3′-UTR and consequently promoted the translocation of YAP1, a key molecule in the Hippo pathway, into the nucleus [89]. MiR-31 is a master regulator of integrins as it targets multiple partners of the α subunits (α2, α5, and αV) of β1 integrins as well as β3 integrins, inhibiting cell proliferation in a ligand-dependent manner [90].
miR-155	↓	Targets are MLH1, MSH2, and MSH6 [91].
miR-182	↑	Promotes cancer cell migration and invasion by inhibiting MBNL2 expression [92].
miR-214	↑	Targets are PTEN/AKT, β-catenin, and tyrosine kinase receptor pathways. MiR-214 also regulates the levels of key modulators of gene expression: the epigenetic repressor Ezh2, p53 ”genome guardian”, the transcription factors TFAP2 and another miRNA, miR-148b. Thus, miR-214 seems to play an important role in coordinating tumor proliferation, stem, angiogenesis, invasiveness, extravasation, metastasis, chemoresistance, and microenvironment [93].
**Pre-metastatic niche formation**		
miR-19a	↑	Member of the highly conservative miR-17-92 cluster [94]. The miR-17/miR-20a seed family is responsible for this anti-aging activity [95]. MiR-19a activates the mammalian protein kinase B (AKT) rapamycin (mTOR) pathway, thereby functionally antagonizing *PTEN* to promote cell survival [96].
miR-126	↑	The target is *VEGF*. It increases the rate of migration and invasion of EC cells [80].
miR-133a	↓	MiR-133a is a suppressor. The miR-1/133a cluster directly regulates *PDE7A* in EC cells. PDEs are enzymes that regulate the cellular levels of cAMP and cGMP second messengers by controlling their rate of degradation [97].
miR-503	↓	It plays a tumor suppressor role by targeting CCND1 [98].

↑ Upregulation; ↓ Downregulation.

**Table 2 ijms-24-11489-t002:** MiRNA changes in obese women.

Upregulation	miR-17 [112] miR-152 [112] miR-205 [113] miR-376a [114] miR-548ag [115]
Downregulation	miR-15b [116] miR-17 [117] miR-138 [112] miR-150 [118] miR-593 [112]

**Table 3 ijms-24-11489-t003:** MicroRNAs changes in metastatic lymph nodes.

Upregulation	miR-21 [128] miR-107-5p [68] miR-429 [129,130] miR-501 [131] miR-576-5p [132]
Downregulation	miR-24b-5p [133] miR-26a-5p [82] miR-34a [65,66] miR-34b-5p [134] miR-34c-3p [134] miR-34c-5p [134] miR-148b [71] miR-204-5p [126] miR-505 [135]

**Table 4 ijms-24-11489-t004:** MicroRNA changes occurring in patients with good prognosis, with long PFS (progression-free survival).

Upregulation	miR-29b [139] miR-126 [81] miR-148b [71,72] miR-152 [140,141] miR-199a-5p [133] miR-214-3p [74] miR-340-5p [77] miR-455-5p [133] miR-505 [135]
Downregulation	miR-429 [129]

**Table 5 ijms-24-11489-t005:** Serum miRNA changes in patients with endometrial cancer.

Upregulation	miR-15a-5p [143] miR-20b-5p [144] miR-27a [145] miR-106b-5p [143] miR-107 [143] miR-143 [44,144] miR-143-3p [144] miR-150-5p [145] miR-186 [146,147] miR-195-5p [144] miR-200a [146] miR-203 [146] miR-204 [146] miR-204-5p [144] miR-222 [146,147] miR-223 [146,147] miR-423-3p [144] miR-449 [146] miR-484 [144] miR-887-5p [148]
Downregulation	miR-16 [44] miR-99b [44] miR-125 [44] miR-145 [44] miR-204 [147]

## Data Availability

Not applicable.

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
