# Peer review of "The Role of miRNAs in the Development, Proliferation, and Progression of Endometrial Cancer"

_ijms, 2023, doi:10.3390/ijms241411489_

Round 1

Reviewer 1 Report

In the current form, the manuscript describes the status of many microRNAs in endometrial cancer or some risk factors patients.

I encourage the author to analyze deeply the role of specific microRNAs, which have been associated with altering pathways related to the disease. Section 3, "Overview of selected microRNAs shows a good approach of the authors to describe the role of microRNAs in endometrial cancer. However, this section keeps away some other interesting microRNAs that must be addressed. Besides this general overview, I have some minor comments listed below.

Line 109. miR-181a is mentioned twice.

Figure 1 needs to be improved.

Author Response

Responses are sent in the attached file.

Reviewer 2 Report

In this article, the authors describe selected miRNAs based on the literature and their role in the development of endometrial cancer. The manuscript is straightforward, well written, and concise and has clear results within the scope of a review article. Definitely deserves to be published and is a valuable contribution to the “International Journal of Molecular Sciences”. The following minor comments need to be addressed, as per my recommendations.

[1]1.2 MicroRNAs”, Page 3 of 28, Lines 98-99:

Therefore, they are regulators of a complex network of processes occurring in the tumor microenvironment [24].”.

Within this context, the authors are recommended to mention their crucial role in the pathogenesis of germ cell tumours (GCT). Lethal-7 (let-7) is a group of nine miRNA that function as important tumour-suppressor genes. Let-7 is negatively regulated by the RNA-binding protein LIN28, which controls the pluripotency of embryonic stem cells. LIN28-positive GCT have been shown to have reduced levels of let-7 miRNA, therefore suggesting that the LIN28/let-7 pathway could have a significant role in the pathogenesis of GCT.

Recommended reference: Cheung A, et al. Non-Epithelial Ovarian Cancers: How Much Do We Really Know? Int J Environ Res Public Health. 2022;19(3):1106.

[2] “1.2 MicroRNAs”, Page 3 of 28, Lines 120-121:

“Currently, miRNAs are attractive candidates for therapeutic targets in the treatment of malignancies. Therefore, identifying their targets is essential for cancer research.”.

At the same time, miRNAs have been found to induce chemoresistance. The authors should mention the example of FOLFOX-resistance in advanced colorectal cancer, which is significantly associated with upregulation and downregulation of several serum miRNAs. The differentiation of FOLFOX-resistant from FOLFOX responsive patients by serum miR-19a had a reported sensitivity and specificity of 66.7 and 63.9%, respectively. In terms of treatment response to anti-VEGF or anti-EGFR inhibitors in metastatic colorectal cancer, upregulation of miR-126 was correlated with bevacizumab resistance, whereas overexpression of miR-31, miR-100, miR-125b, and downregulation of miR-7, with resistance to cetuximab, respectively.

Recommended reference: Boussios S, et al. The Developing Story of Predictive Biomarkers in Colorectal Cancer. J Pers Med. 2019;9(1):12. 

Good level of English

Author Response

(The authors gave the same response as above.)

Round 2

Reviewer 1 Report

The authors have addressed my comments accordingly.